# CT-Based Radiomics and Deep Learning for BRCA Mutation and Progression-Free Survival Prediction in Ovarian Cancer Using a Multicentric Dataset

**DOI:** 10.3390/cancers14112739

**Published:** 2022-05-31

**Authors:** Giacomo Avesani, Huong Elena Tran, Giulio Cammarata, Francesca Botta, Sara Raimondi, Luca Russo, Salvatore Persiani, Matteo Bonatti, Tiziana Tagliaferri, Miriam Dolciami, Veronica Celli, Luca Boldrini, Jacopo Lenkowicz, Paola Pricolo, Federica Tomao, Stefania Maria Rita Rizzo, Nicoletta Colombo, Lucia Manganaro, Anna Fagotti, Giovanni Scambia, Benedetta Gui, Riccardo Manfredi

**Affiliations:** 1Department of Bioimaging, Radiation Oncology and Hematology, Fondazione Policlinico Universitario Agostino Gemelli IRCCS, Largo A. Gemelli 8, 00168 Rome, Italy; giacomo.avesani@policlinicogemelli.it (G.A.); huongelena.tran@policlinicogemelli.it (H.E.T.); luca.russo@guest.policlinicogemelli.it (L.R.); salvatorepersiani@gmail.com (S.P.); miriam.dolciami@guest.policlinicogemelli.it (M.D.); luca.boldrini@policlinicogemelli.it (L.B.); jacopo.lenkowicz@policlinicogemelli.it (J.L.); riccardo.manfredi@policlinicogemelli.it (R.M.); 2Department of Experimental Oncology, IEO European Institute of Oncology IRCCS, Via Ripamonti 435, 20141 Milan, Italy; giulio.cammarata@gmail.com (G.C.); sara.raimondi@ieo.it (S.R.); 3Medical Physics Unit, IEO European Institute of Oncology IRCCS, Via Ripamonti 435, 20141 Milan, Italy; francesca.botta@ieo.it; 4Radiology Unit, Ospedale Centrale di Bolzano, Via Lorenz Böhler, 5, 39100 Bolzano, Italy; matteo.bonatti@sabes.it; 5Gynecology and Obstetrics Unit, Ospedale Centrale di Bolzano, Via Lorenz Böhler, 5, 39100 Bolzano, Italy; tiziana.tagliaferri@sabes.it; 6Department of Radiological, Oncological and Pathological Sciences, “Sapienza” University of Rome, Piazzale Aldo Moro, 5, 00185 Rome, Italy; veronica.celli@uniroma1.it (V.C.); lucia.manganaro@uniroma1.it (L.M.); 7Division of Radiology, European Institute of Oncology IRCCS, Via Ripamonti 435, 20141 Milan, Italy; paola.pricolo@ieo.it; 8Department of Maternal and Child Health and Urological Sciences, “Sapienza” University of Rome, Piazzale Aldo Moro, 5, 00185 Rome, Italy; federica.tomao@uniroma1.it; 9Clinica di Radiologia EOC, Istituto Imaging della Svizzera Italiana (IIMSI), Via Tesserete 46, 6900 Lugano, Switzerland; stefaniamariarita.rizzo@eoc.ch; 10Facoltà di Scienze Biomediche, Università della Svizzera Italiana (USI), Via Giuseppe Buffi 13, 6900 Lugano, Switzerland; 11Medical Gynecologic Oncology Unit, European Institute of Oncology IRCCS, Via Ripamonti 435, 20141 Milan, Italy; nicoletta.colombo@ieo.it; 12University of Milan-Bicocca, Piazza dell’Ateneo Nuovo, 1, 20126 Milan, Italy; 13Department of Woman, Child and Public Health, Fondazione Policlinico Universitario Agostino Gemelli IRCCS, Largo A. Gemelli 8, 00168 Rome, Italy; anna.fagotti@policlinicogemelli.it (A.F.); giavanni.scambia@policlinicogemelli.it (G.S.); 14Catholic University of the Sacred Hearth, Largo Francesco Vito, 1, 00168 Rome, Italy

**Keywords:** ovarian cancer, radiomics, computed tomography, machine learning

## Abstract

**Simple Summary:**

Ovarian cancer has a heterogeneous response to treatment, and relapse may vary considerably. Different studies investigated the role of radiomics in ovarian cancer. However, many of them were performed in a single center, and solid external validation of findings is still missing. We used a multicentric database of high-grade serous ovarian cancer to build predictive radiomic and deep-learning models for early relapse and BRCA mutation, validating them in a different set of cases coming from other institutions. In our multicentric dataset, representative of a real-life clinical scenario, we could not find a good radiomic predicting model for PFS and BRCA mutational status with both traditional radiomics and deep learning methods. This study highlights that to implement the radiomics approach in clinical routine, we still need standardization of acquisition protocols, validation of harmonization method and radiomic pipelines, other than robust, prospective, multicentric, external validations of findings.

**Abstract:**

Purpose: Build predictive radiomic models for early relapse and BRCA mutation based on a multicentric database of high-grade serous ovarian cancer (HGSOC) and validate them in a test set coming from different institutions. Methods: Preoperative CTs of patients with HGSOC treated at four referral centers were retrospectively acquired and manually segmented. Hand-crafted features and deep radiomics features were extracted respectively by dedicated software (MODDICOM) and a dedicated convolutional neural network (CNN). Features were selected with and without prior harmonization (ComBat harmonization), and models were built using different machine learning algorithms, including clinical variables. Results: We included 218 patients. Radiomic models showed low performance in predicting both BRCA mutation (AUC in test set between 0.46 and 0.59) and 1-year relapse (AUC in test set between 0.46 and 0.56); deep learning models demonstrated similar results (AUC in the test of 0.48 for BRCA and 0.50 for relapse). The inclusion of clinical variables improved the performance of the radiomic models to predict BRCA mutation (AUC in the test set of 0.74). Conclusions: In our multicentric dataset, representative of a real-life clinical scenario, we could not find a good radiomic predicting model for PFS and BRCA mutational status, with both traditional radiomics and deep learning, but the combination of clinical and radiomic models improved model performance for the prediction of BRCA mutation. These findings highlight the need for standardization through the whole radiomic pipelines and robust multicentric external validations of results.

## 1. Introduction

Ovarian cancer (OC) is the fifth deadliest cancer among women [1], and high-grade serous epithelial cancer is the most common histological subtype [2]. Magnetic resonance imaging (MRI) is the best technique to distinguish between benign and malignant lesions, and radiomics is a field of research in this setting [3]. Still, CT is the preoperative staging technique for OC following European Society of Urogenital Radiology (ESUR) guidelines [4]. Standard treatment for OC is primary cytoreductive surgery combined with platinum-based chemotherapy [5], independently from individual prognostic factors. The follow-up scheme is also similar for all patients [6]. The main genetic alteration considered is BRCA 1–2 mutation, associated with a better prognosis [7] and may benefit from introducing PARP-inhibitor drugs in maintenance therapy [8]. However, OC has a heterogeneous response to treatment, and relapse time may vary considerably [9]. In this context, it may be critical to identify patients at high risk of recurrence.

Radiomics is an innovative method to extract quantitative data about microscopic characteristics of tissues from clinical images; these data can be combined with clinical, genomic, proteomic and other information to build new models and biomarkers to predict specific targets, such as diagnosis, treatment response, survival or genomic and proteomic alterations [10]. In the classical radiomics approach, the “hand-crafted features” derive from predefined mathematical formulas based on intensity, shape and texture of a region of interest identified in radiological images [11]. The radiomic features are then fed into machine learning models, mainly random forests and support vector machine [12]. Deep learning has allowed the extraction of “deep features” using convolutional neural networks (CNN) without any predefined external imposition, investigating a much higher number of features and opening new horizons for image analysis. Describing radiomics and deep learning methods and pipelines is beyond the scope of this paper, but different reviews are available in the literature.

Recently, some studies have investigated the radiomic approach in ovarian cancer, discovering a correlation with metastatic [13] and lymph node [14] involvement using regularized logistic regression models, residual tumor after surgery by means of Kaplan-Meier analysis [15], progression-free survival (PFS) using Cox proportional hazards and logistic regression models [15,16,17,18,19], overall survival (OS) [20], genetic mutations including BRCA [17,21] and proteomic profile [22] of the tumor. However, almost all these studies were performed in a single center, many had small datasets, and robust external validation of findings is still missing.

Since radiomic features can be affected by variations among scanners and acquisition parameters, different harmonization methods have been proposed in the literature to obtain more reproducible radiomic features. The harmonization methods can be classified into image domain and feature domain approaches [23]. ComBat harmonization is a feature domain method, initially proposed in genomics, that has recently been applied in radiomics [24,25]. ComBat method was specifically developed to address the so-called batch effect, which is typical in genomics and refers to the generic difficulties of analyzing samples derived from different laboratories. Similarly, the numerical value of radiomic features conveys different information, as they are affected by the variation in image acquisition parameters. By realigning feature values among centers, ComBat harmonization allows for the pooling of data from different centers without losing statistical power caused by center variability.

Our study aims to create a multicentric database of high-grade serous ovarian cancer to build predictive radiomic models for early relapse and BRCA mutation and to validate them in a different set of cases coming from different institutions. We also wanted to investigate a deep learning approach for the same outcomes.

## 2. Materials and Methods

### 2.1. Patient Selection

The first group of patients came from IEO, Milan (Group 1); the study population was previously investigated in Rizzo et al. [18].

We collected patients from three Italian centers: Fondazione Policlinico Gemelli, Rome (Group 2);Policlinico Umberto I, Rome (Group 3);Ospedale Centrale, Bolzano (Group 4).

The inclusion criteria were:−Availability of a pre-treatment contrast-enhanced CT study of at least abdomen and pelvis in portal-venous phase;−Surgery for staging or complete debulking;−Diagnosis of high-grade serous ovarian cancer.

Exclusion criteria were:−CT slice thickness >5 mm;−Surgery performed in another center;−Other histologies of ovarian cancer;−Previous history of malignancy.

### 2.2. Image Acquisition

We retrospectively extracted DICOM files of preoperative CT scan from each center’s respective Picture Archiving and Communication System (PACS). We considered only the post-contrast portal-venous phase. All images had a slice thickness of 1–5 mm, and all manufacturers (GE Medical Systems, Siemens, Philips, Toshiba, Hitachi) were included. 

All participant centers are referral hospitals highly specialized for the treatment of ovarian cancer; not all exams were performed in those centers, but most of them were collected from many different smaller hospitals where the diagnosis was performed; when CTs were diagnostic and were not older than two weeks, scans were not repeated. Because of that, the acquisition parameters differed among CTs even in the same center. We decided not to select exams based on acquisition parameters to reflect the real-life scenario. However, we included only those exams where soft tissue kernel reconstruction was unavailable. 

### 2.3. Clinical Data 

Clinical data were age, preoperative CA-125, FIGO stage, BRCA status and PFS, defined as radiologic or biochemical progression.

### 2.4. Image Segmentation

For Group 1, the image segmentation methodology was described in the previous paper [18]. 

For Groups 2, 3 and 4, manual segmentation of the entire primary tumor (Gross Tumor Volume, GTV) for each patient was performed using ITK-Snap [26] (Figure 1). 

For all groups, GTV was the abnormal ovary or the mass in the pelvis when ovaries were no more distinguishable. The segmentation was drawn on each slice where the tumor was present to include the whole tumor with particular attention to excluding vascular and intestinal structures. The segmentations were performed by two radiologists with three years of experience in gynecologic radiology; all segmentations were checked by a third radiologist with ten years of experience in gynecologic imaging and with specific expertise in segmentation in radiomic studies.

### 2.5. Study Design

We developed models for predicting 1-year PFS and BRCA mutation from hand-crafted radiomic features or deep learning features. In particular, the hand-crafted radiomic features were fed into machine learning models, while deep learning features were extracted using CNN to develop a deep learning model. For the classification of 1-year PFS, we denoted patients with 1-year relapse as belonging to class 1 (cases) and patients without 1-year relapse as belonging to class 0 (controls). For the classification of BRCA mutation, we denoted patients with BRCA 1–2 mutations as belonging to class 1 (cases) and patients without BRCA 1–2 mutations or with variants of uncertain significance (VUS) as belonging to class 0 (controls).

The training set used to train the models included the patients from Group 1 and Group 2. The testing set used for external validation included the patients from Groups 3 and 4. 

### 2.6. Radiomic Feature Extraction and Analysis

Radiomic and statistical analyses were performed in RStudio (R version 4.4.1) and using Python 3.7. 

Handcrafted radiomic features were extracted for all groups from the GTV using an open-source R library called MODDICOM (https://github.com/kbolab/moddicom, accessed on 29 March 2022), implemented for radiomic features extraction by the Radiomics Research Core facility of the Fondazione Policlinico Universitario “A. Gemelli” IRCCS, Rome, Italy [27]. The computation of the radiomic features is fully compliant with the recommendations of the Image Biomarker Standardization Initiative [28]. 

A total of 217 radiomic features were extracted for each GTV. Features belonged to three different families: statistical, morphological, and textural features.

To evaluate feature heterogeneity among centers, an analysis of reproducibility of the radiomic features against the image acquisition parameter given by the slice thickness and the CT scanner manufacturer was performed using a two-way ANOVA. This analysis excluded unreproducible features that presented significantly different means due to different slice thickness and manufacturer (false discovery rate corrected *p* value < 0.05). The smaller set of reproducible features was used for the first analysis.

Moreover, as a sensitivity analysis, the ComBat harmonization [29] method was applied to all 217 radiomic features using the *ez.combat* R package, available on the Comprehensive R Archive Network (CRAN). Then, a new two-way ANOVA was performed, and no unstable features were found. This allowed us to use all features for additional analysis and compare it with the first analysis based on reproducible features only. 

Following the reproducibility analysis or the sensitivity analysis, we conducted a further feature selection on the training set to reduce the number of radiomic features included in the modeling process and prevent overfitting. First, we performed a univariate analysis with the Wilcoxon–Mann–Whitney statistical test to consider only features that showed a statistically significant difference in the two classes (i.e., 1-year PFS as yes/no and BRCA mutation as present/absent), with a significance level set to 0.05. Then, we removed the multicollinearity between the features included in the models by computing the Pearson cross-correlation coefficient. We set a threshold of 0.9 to exclude highly correlated features. 

### 2.7. Radiomic Models

Radiomic models were obtained using radiomic features only. In addition, clinical–radiomic models were generated combining radiomic features and relevant clinical variables. For the classification of 1-year PFS, we included age, family history of ovarian or breast cancer, residual tumor. For the classification of BRCA mutation, we considered age and family history of ovarian or breast cancer.

Different machine learning models were developed using the radiomic features and clinical data of the training set in Python 3.7. These models include a penalized logistic regression with L2 penalty, random forest, Support Vector Machine (SVM) implemented in the standard *scikit-learn* package, XGBoost using the *XGBoost* package. Fine tuning of the hyper-parameters was performed with a randomized grid search using a 5-fold cross-validation (CV).

For each developed model, we computed the area under the curve (AUC) of the receiver operating characteristic (ROC) in each of the five CV folds. We computed the training set’s mean AUCs over the five CV folds. Then, the models were applied to the external validation cohort and the AUC of the ROC was computed.

### 2.8. Deep Learning Model 

As a second approach, we adopted the 2D-CNN built by Lombardo et al. [30], which showed high performance in tumor classification. Image pre-processing was conducted in Python 3.7, while the 2D-CNN was implemented in Tensorflow 2.4.0. Training and testing were performed on a single graphics processing unit (GPU), a NVIDIA Quadro RTX 5000 with 16 GB of memory.

The CNN is defined by three 2D-convolutional blocks, two fully connected layers, one dropout layer and an output layer. Every convolutional block consists of a convolutional layer, a max-pooling layer and a parametric rectified linear unit (PReLu) activation function. 

Unlike the 2D-CNN experiment in Lombardo et al., which used as input the axial slice with the highest number of tumor voxels, we decided to use every axial slice presenting tumor voxels as input. Before data augmentation, CT images were isotropically resampled to a 1 × 1 × 1 mm^3^ grid using linear interpolation and masked using binary masks isotropically resampled to a 1 × 1 × 1 mm^3^ grid using nearest neighbor interpolation. The re-sampled masked CTs cropped to a dimension of 256 × 256 were used as input to the networks. 

Data augmentation was performed by applying random cropping with a central random shift and reducing the images to 128 × 128. 

For the 1-year relapse classification, the learning rate was set at 10-3 and batch size at 64. For the BRCA classification, the learning rate was set at 10-3 and batch size at 16. We imposed weight decay at 10-4 and dropout rate at 25% for both endpoints to reduce overfitting. Networks weights and biases were optimized using the Adam algorithm [31]. The loss function used was the binary cross-entropy. Hyper-parameters were searched using a manual grid search and were chosen as the ones providing the highest 5-fold cross-validation mean AUC. The CNN was trained for 200 epochs with early stopping by setting a patience of 50 epochs. 

The five best models from the 5-fold cross-validation were applied to the external validation cohort for each classification problem. We averaged the five predictions resulting from the five best models to obtain a single prediction for each slice. We performed bootstrap resampling of the slices of the patients in the test set containing the tumor with 100 iterations and computed the median AUC.

## 3. Results

A total of 218 patients were recruited. Clinical data are summarized in Table 1.

For the prediction of 1-year relapse, the training set for building the models presented a proportion of 34% for 1-year relapse, while the test set for external validation showed a ratio of 26% (*p* > 0.05).

The patients with missing information regarding the BRCA mutation were excluded from the analysis. For the prediction of the BRCA mutation, the training set for building the models presented a proportion of 38% for the BRCA mutation, while the test set for external validation showed a ratio of 40% (*p* > 0.05). 

### 3.1. Radiomic Feature Analysis and Selection

The reproducibility analysis performed with the ANOVA on the 217 original radiomic features found 71 reproducible features. ComBat was applied to 217 patients due to missing information about the manufacturer for one patient. The analysis of reproducibility performed with ANOVA on the 217 harmonized features obtained from the application of ComBat showed that all 217 features were reproducible (Figure 2A,B).

For the 1-year relapse classification problem following ANOVA without harmonization, the univariate analysis retained 16 features, while the final set of selected uncorrelated features included 6 features belonging to the statistical and textural families (namely ‘F_stat.kurt’, ‘F_stat.min’, ‘F_cm_merged.joint.max’, ‘F_rlm.r.perc’, ‘F_szm.lze’, ‘F_szm.z.perc’). 

For the 1-year relapse classification problem based on the features harmonized with ComBat, the univariate analysis retained 83 features, while the final set of selected uncorrelated features included 17 features belonging to the statistical and textural families (namely ‘F_stat.min’, ‘F_stat.max’, ‘F_stat.range’, ‘F_cm.diff.avg’, ‘F_cm.diff.entr’, ‘F_cm.inv.diff.norm’, ‘F_cm.inv.var’, ‘F_cm.corr’, ‘F_cm_merged.joint.max’, ‘F_cm_merged.joint.entr’, ‘F_cm_2.5D.joint.entr’, ‘F_cm_2.5D.energy’, ‘F_cm_2.5D.corr’, ‘F_cm.2.5Dmerged.info.corr.1′, ‘F_rlm.glnu.norm’, ‘F_rlm.r.perc’, ‘F_szm.z.perc’).

For the BRCA classification problem following ANOVA without harmonization, the univariate analysis retained six features, while the final set of selected uncorrelated features included three textural features (namely ‘F_rlm.r.perc’, ‘F_szm.lze’, ‘F_zsm.z.perc’). 

For the BRCA classification problem based on the features harmonized with ComBat, the univariate analysis retained 59 features, while the final set of selected uncorrelated features included eight textural features (namely ‘F_cm.diff.avg’, ‘F_cm.diff.entr’, ‘F_cm.inv.var’, ‘F_cm.corr’, ‘F_cm_2.5D.corr’, ‘F_rlm.r.perc’, ‘F_szm.lzhge’, ‘F_szm.z.perc’).

The list of names of the selected radiomic features is available in the Appendix A.

### 3.2. Radiomics and Deep Learning Models 

For the 1-year relapse classification, radiomics and clinical–radiomic models were trained on 218 patients and six and nine features using non-harmonized data, respectively. In general, models performed poorly on training and test sets. For the radiomic models, the highest five-fold cross-validation AUC was estimated as 0.63 using XGBoost, while the highest test set AUC was estimated as 0.56 using Random Forest and XGBoost models (Table 2). The model performance did not improve when including the clinical variables in the clinical–radiomic models (Table 3). When applied to harmonized data (217 patients, 17 and 20 features for the radiomic and clinical–radiomic models, respectively), the model’s performances were generally slightly worse when compared to the non-harmonized data (Table 2 and Table 3).

For the 1-year relapse classification, the deep learning model was trained using 9849 axial slices and tested with 5473 axial slices. Similar to radiomics models, the 2D CNN performed had low performance on the training set (average five-fold CV AUC = 0.60) and test set (AUC = 0.50) (Table 2). 

When investigating BRCA classification, the radiomic models built with non-harmonized data (218 patients, 3 features) produced the highest training set AUC as 0.62 using XGBoost, while the highest test set AUC was 0.59 using SVM (Table 4). In this case, the inclusion of the clinical variables in the clinical–radiomic models (218 patients, 5 features) improved the model performance up to an AUC of 0.75 with the XGBoost in the training set and AUC of 0.74 with the penalized logistic regression in the test set (Table 5). Figure 3 shows the ROC curve obtained for the best-performing model obtained for the test set. When applied to harmonized data (217 patients, 8 and 10 features for the radiomic and clinical–radiomic models, respectively), Random Forest provided the highest AUC both in training and test sets for the radiomic models (AUC = 0.65 and AUC = 0.50, respectively), while the model performance improved for the clinical–radiomic models with the highest AUC of 0.76 obtained in training set with XGBoost, and the highest AUC of 0.70 with SVM in the test set (Table 4 and Table 5).

For BRCA classification, the deep learning model was trained using 9589 axial slices and tested with 3417 axial slices. Similar to the 1-year relapse endpoint, the 2D-CNN did not provide an accurate prediction, providing a training set AUC of 0.56 and a test set AUC of 0.48. (Table 4).

## 4. Discussion

Radiomics may be seen as a promising tool for personalized medicine. It is non-invasive and relies on medical images as a source of data already part of clinical practice. Recently many studies investigated the possibility of radiomics being used to identify biomarkers in ovarian cancer.

To overcome some limitations of previous studies, we created and used a multicentric database: this enables us to enlarge the dataset but also gives the possibility to train the algorithms on real-world data that comprises many different settings that may affect imaging and features reproducibility; moreover, the multicentric dataset permits to test the developed models in external cohorts, which is a fundamental step to generalize findings.

We chose to perform the analysis on CTs because it is the most common imaging method to stage advanced ovarian cancer. We considered only the post-contrast portal-venous phase because it is the most commonly performed phase for ovarian cancer staging. Moreover, the signal intensities of CT data given by the Hounsfield units are intrinsically quantitative and not operator-dependent, thus requiring less complex standardization than compared to MRI and (US) ultrasound (US) [32,33].

The two outcomes, BRCA mutational status and PFS, were chosen because they are significant and widely used in clinical practice. BRCA testing is recommended in clinical practice because patients with that genetic alteration had a better response to platinum-based therapy and are targetable with PARP-inhibitor agents [7,8]. PFS was chosen as outcome because identifying those patients with a higher risk of relapse may imply a stricter follow-up and more prompt intervention [34]. 

We included in the analysis the GTV because ovarian cancer is highly heterogeneous and we wanted to include all information contained in the images. This is in accordance with previous studies that tested radiomic analysis in survival and BRCA prediction. The inclusion of GTV differed from the original research the CNN comes from. In that study [30], the single slice with the most extensive area of the tumor was used for the 2D-CNN experiments. We decided to include all slices of the GTV also to develop the deep learning model because ovarian masses often have cystic components and considering a single slice may lead to losing many relevant pieces of information. We did not implement a 3D-CNN due to its computational demand.

In Rizzo et al. [18], which included part of the patients used in the present study, the addition of a radiomic feature to a clinical model improved the performance in predicting early recurrence. We used the same outcome (dichotomized 12-month PFS). Still, we performed a different analysis. First, we extracted the radiomic features again with new software (MODDICOM) from all GTVs (including Group 1) to extract the same radiomic features from all groups. Moreover, we added another large group (Group 2) of patients to the training set because our aim was not to validate a single study but to create a new model on a larger multicentric dataset. 

Meier et al. [17] included 88 patients with high-grade serous ovarian cancer and found a correlation between texture characteristics of ovarian cancer with PFS. Different from the present study, they investigated heterogeneity between different tumor sites and not the primary tumor characteristics. The heterogeneity between different tumor locations probably reflects the diverse biological characteristics of metastasis, which permits the tumor to elude the effect of treatments.

Zargari et al. [35] used as outcome for chemotherapy response the PFS at six months; they dichotomized the outcome as well but chose a different temporal threshold. They included all tumor volume, including primary and metastatic localizations. They included 103 patients; a correlation between the radiomics model and 6-month PFS was found; however, they had a monocentric dataset and the predictive value was calculated only by cross-folded validation. An interesting aspect of this paper is that spatial and frequency features have almost the same weight in the final model. Our work also included those kinds of features.

Wei et al. [16] created a nomogram based on radiomic signature and clinical data that demonstrated good performance in predicting PFS; they included 142 patients and chose as thresholds for PFS 18 and 36 months. This was a multicentric study with validation and test sets. In addition, in this study, they used four different scanners in two institutions but, contrary to our results, they stated that this had a small impact on the model and results. Its main limitation is the lack of correlation with the biological interpretation of findings which were however promising.

Chen et al. [36] also created a mixed clinical–radiomic model that overperformed the clinical model in predicting PFS; their sample size was quite big (256 patients), but the dataset was monocentric and the model’s overall performance was moderate (AUC = 0.77). In our experience, combining radiomic and clinical features improved the results of the models. 

The biggest study about radiomics predicting PFS was recently published by Fotopoulou et al. [15]; this was an external validation of a previously published Radiomic Prognostic Vector (RPV) [20] which was associated with short OS; the RPV was associated with the stromal component of the tumor and inversely with DNA damage response. RPV had already been validated in an external cohort of patients and a biological interpretation based on genomic data was given. The new external validation confirmed the role of RPV but the significant outcome was slightly different in the two studies (OS vs. PFS).

Regarding BRCA, to the best of our knowledge, the only study that proposed a radiomic model with good performance (AUC = 0.82) in predicting BRCA status is that by Mingzhu et al. [21]. They included different phases of post-contrast CTs in the analysis, enlarging the set of possibly correlated features. The included patients were 106; the dataset was monocentric and there was only internal validation.

Using real-world data, our analysis failed to identify a reliable radiomics model that predicts both early recurrence and BRCA mutation. The performance of both radiomic and deep learning models showed low discriminative performance even in the training set, raising concern about the high noise level contained in the dataset. This may depend on the composition of the dataset: the two centers included in the training set are referral centers for ovarian cancer. Many patients already had a staging CT when they arrived at these centers. The included CTs derived from many different smaller centers and image acquisition procedures may have varied widely, affecting radiomics analysis. The acquisition is the first and fundamental step in radiomic analysis [37] and its variation may affect the downstream pipeline. This is testified by the high number of features excluded because they are unstable and would be in accordance with previous findings [38]. In Rizzo et al. [18], many features were excluded because of reproducibility and redundancy. In addition, other image acquisition parameters or reconstruction settings that were not considered in this study may affect the reproducibility of radiomic features among different centers.

We decided to introduce ComBat harmonization before feature selection and machine learning modeling to mitigate features’ lack of reproducibility given to differences among centers. However, the performance did not improve, suggesting that the standard ComBat method could not harmonize the data among the different centers based on the information provided by slice thickness and manufacturer.

For radiomics models, we eliminated highly correlated features as commonly performed when developing predictive models in radiomics studies to reduce overfitting. We trained the models considering all reproducible features to understand if a too restrictive feature selection caused the model’s poor performances. However, as expected, performances were even worse (not shown in the results section), including collinear and redundant features.

The combination of radiomic and clinical variables produced clinical–radiomic models that improved model performance to predict BRCA mutation. We obtained promising results with an AUC of 0.74 for the testing set when using non-harmonized data.

Using a different statistical methodology, we also applied deep learning to test if artificial intelligence would overcome the difficulties of the traditional radiomic approach. In classification analysis, Sun et al. [39] found that deep learning provided slightly better results when predicting axillary lymph node metastasis. For survival analysis, Wang et al. [40] extracted deep learning features using a deep learning network and used them as input for a Cox-PH model. They found a moderate performance of the deep learning model in predicting 3-year PFS. Different from theirs, our deep learning model did not overperform the other traditional models. Our deep learning approach was exploratory and we adopted the simple CNN architecture from Lombardo et al. [30] to minimize the number of model parameters and reduce overfitting due to the relatively small sample size available for training. Transfer learning strategies may help addressing the problem of limited data for deep learning models [41]. Furthermore, the same issues of image standardization may have affected both traditional radiomic models and deep learning. 

Our findings underline the limitations of radiomics: reproducibility of features and models among different centers is low, affected by many technical variables that may change between different scanners; the reproducibility of findings of different studies is low too, lacking in literature a real independent validation of any published model; Furthermore, the biological interpretation of radiomics findings is weak in many studies on this field, and recent works suggested that biological correlation with radiomic features is not mandatory [42]. Standardization protocols were proposed to overcome these limits of the radiomic approach, such as IBSI [28], but probably standardization of other pipeline steps is necessary. Nevertheless, our study showed that the inclusion of relevant clinical variables could improve model performance in predicting BRCA mutation.

A limitation of this study is its retrospective nature, with all possible selection bias. Another limitation is the heterogeneity of the included images, which, conversely, reflects the real-world application of radiomics studies. A different harmonization method than the ComBat harmonization used in this study may improve the reproducibility of the radiomic features and produce radiomic models able to predict the considered outcomes [23]. In addition, feature selection was performed on the whole training set, while incorporating the feature selection process in each fold of the cross-validation may have better tested the generalization capabilities of the optimized hyperparameters. We did not include any explainability method about the deep learning method. Moreover, we included in the analysis only primary tumors, while other recent papers [19,22,43,44] suggested the importance of inter-site tumor heterogeneity in predicting ovarian cancer behavior. 

## 5. Conclusions

In our multicentric dataset, representative of a real-life clinical scenario, we could not find a good radiomic predicting model for PFS and BRCA mutational status with both traditional and deep radiomics. However, the inclusion of relevant clinical variables in combined clinical–radiomic models slightly improved model performance for the prediction of BRCA mutation. This study highlights that to implement radiomics approaches in clinical routine, we still need standardization of acquisition protocols, validation of harmonization method and radiomic pipelines, other than robust, prospective, multicentric, external validations of findings.

## Figures and Tables

**Figure 1 cancers-14-02739-f001:**
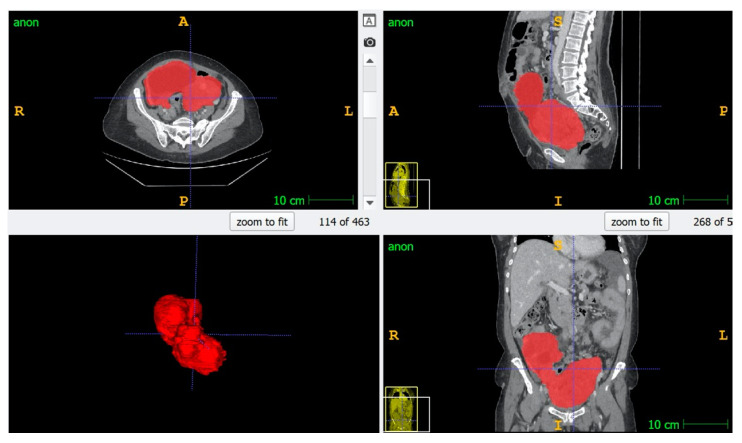
An example of segmentation of tumor volume using ITKsnap.

**Figure 2 cancers-14-02739-f002:**
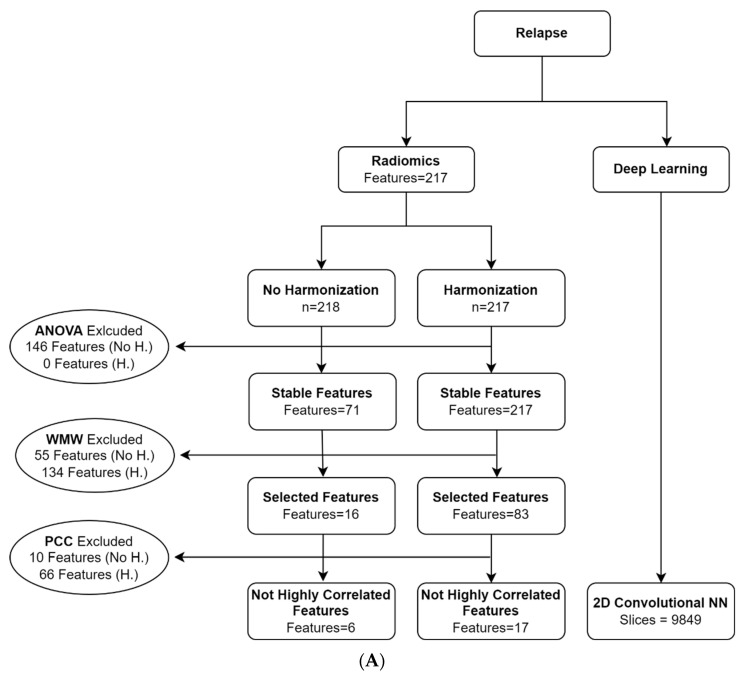
(**A**) Flowchart of the patients and features selections for relapse (*n* = number of patients; Features = number of features; slices = number of included slices; WMW: Wilcoxon–Mann–Whitney; PCC: Pearson cross-correlation; NN: neural network). (**B**) Flowchart of the patients and features selections for BRCA mutation (*n* = number of patients; Features = number of features; slices = number of included slices; WMW: Wilcoxon–Mann–Whitney; PCC: Pearson cross correlation; NN: neural network).

**Figure 3 cancers-14-02739-f003:**
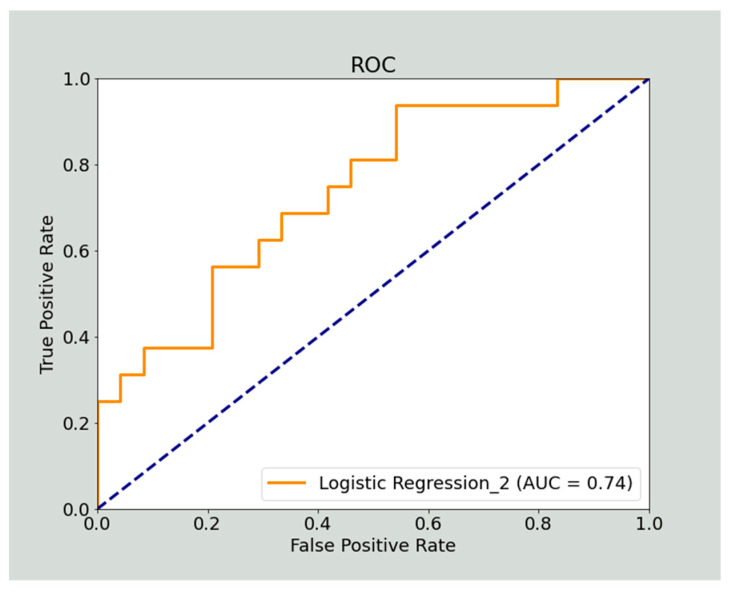
ROC curve of the best performing model given by the clinical–radiomic model for the BRCA mutation prediction, obtained by applying the penalized logistic regression to non-harmonized data.

**Table 1 cancers-14-02739-t001:** Clinical information of included patients.

KERRYPNX	Group 1 (*n* = 101)	Group 2 (*n* = 51)	Group 3 (*n* = 32)	Group 4 (*n* = 34)
Age (Mean; Min-Max)	53; 36–76	58; 41–81	63; 29–86	58; 31–83
Family history of ovarian/breast cancer				
0	49 (48.5%)	31 (60.8%)	25 (78.1%)	49 (48.5%)
1	52 (51.5%)	20 (39.2%)	7 (21.9%)	52 (51.5%)
Pathological stage				
1	0 (0%)	0 (0%)	4 (12.5%)	1 (2.9%)
2	11 (10.9%)	0 (0%)	4 (12.5%)	2 (5.9%)
3	66 (65.3%)	38 (74.5%)	22 (68.8%)	21 (61.8%)
4	24 (23.8%)	9 (17.7%)	2 (6.2%)	10 (29.4%)
NA	0 (0%)	4 (7.8%)	0 (0%)	0 (0%)
Residual tumor				
0	74 (73.3%)	41 (80.4%)	23 (71.9%)	24 (70.6%)
1	27 (26.7%)	10 (19.6%)	9 (28.1%)	14 (29.4%)
BRCA				
0	63 (62.4%)	29 (56.9%)	4 (12.5%)	21 (61.8%)
1	38 (37.6%)	19 (%)	4 (12.5%)	12 (35.3%)
NA	0 (0%)	3 (%)	24 (75%)	1 (2.9%)
Recurrence				
0	58 (57.4%)	41 (80.4%)	29 (90.6%)	20 (58.8%)
1	43 (42.6%)	10 (19.6%)	3 (9.4%)	14 (41.2%)

**Table 2 cancers-14-02739-t002:** Results of the models developed for the 1-year relapse classification. Mean AUC of the 5-fold cross-validation (CV) obtained during model training and AUC of the test set for the analysis without or with ComBat harmonization are reported.

	No Harmonization (ComBat)		Harmonization (ComBat)	
Model	Training Set 5-Fold CV AUC	Test Set AUC	Training Set 5-Fold CV AUC	Test Set AUC
Penalized Logistic Regression	0.56	0.48	0.51	0.46
Random Forest	0.62	0.56	0.60	0.48
XGBoost	0.63	0.56	0.61	0.52
SVM	0.56	0.55	0.56	0.45
2D-CNN	0.61	0.5	-	-

**Table 3 cancers-14-02739-t003:** Results of the clinical–radiomic models developed for the 1-year relapse classification. Mean AUC of the 5-fold cross-validation (CV) obtained during model training and AUC of the test set for the analysis without or with ComBat harmonization are reported.

	No Harmonization (ComBat)		Harmonization (ComBat)	
Model	Training Set 5-Fold CV AUC	Test Set AUC	Training Set 5-Fold CV AUC	Test Set AUC
Penalized Logistic Regression	0.60	0.61	0.53	0.54
Random Forest	0.61	0.58	0.60	0.48
XGBoost	0.64	0.47	0.60	0.51
SVM	0.57	0.62	0.57	0.59

**Table 4 cancers-14-02739-t004:** Results of the models developed for the BRCA classification. Mean AUC of the 5-fold cross-validation (CV) obtained during model training and AUC of the test set for the analysis without or with ComBat harmonization are reported.

	No Harmonization (ComBat)		Harmonization (ComBat)	
Model	Training Set 5-Fold CV AUC	Test Set AUC	Training Set 5-Fold CV AUC	Test Set AUC
Penalized Logistic Regression	0.58	0.57	0.61	0.46
Random Forest	0.61	0.48	0.65	0.50
XGBoost	0.62	0.43	0.64	0.45
SVM	0.61	0.59	0.61	0.46
2D-CNN	0.56	0.48	-	-

**Table 5 cancers-14-02739-t005:** Results of the clinical–radiomic models developed for the BRCA classification. Mean AUC of the 5-fold cross-validation (CV) obtained during model training and AUC of the test set for the analysis without or with ComBat harmonization are reported.

	No Harmonization (ComBat)		Harmonization (ComBat)	
Model	Training Set 5-Fold CV AUC	Test Set AUC	Training Set 5-Fold CV AUC	Test Set AUC
Penalized Logistic Regression	0.70	0.74	0.69	0.67
Random Forest	0.71	0.63	0.73	0.62
XGBoost	0.75	0.60	0.76	0.64
SVM	0.71	0.70	0.64	0.70

## Data Availability

Dataset of images and clinical information can be requested from the corresponding author.

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
