# Peer review of "CT-Based Radiomics and Deep Learning for BRCA Mutation and Progression-Free Survival Prediction in Ovarian Cancer Using a Multicentric Dataset"

_cancers, 2022, doi:10.3390/cancers14112739_

Round 1

Reviewer 1 Report

The manuscript presents an interesting study that tries to predict PFS and BRCA mutation for high grade serous ovarian cancer using radiomics           features. The topic is interesting as there are not many radiomics studies for HGSOC yet. Of note, the authors use a multi-institutional cohort and     test their models externally, which is rarely the case in these types of studies. Their honest reporting of the (negative) external validation results  is to be commended. Their application of a batch effect removal step using ComBat is also very interesting.

 The analysis has two important limitations which I think could be addressed before concluding that there is no signal. First, even though important     clinical features such as age, family history or stage were collected, for some reason they were not used in the models. Second, it seems that in the   optimization process the validation data was included in the feature selection, which may have affected the performance.

 Detailed comments below:

 1. Feature selection is performed on the whole training set, which is then also used for hyperparameter optimization using cross-validation. This       means that the CV estimate is probably an over-estimate, and that the optimization process itself is not really testing the generalization              capabilities of the chosen hyperparameter combinations. Either this limitation should be acknowledged (which would leave the external validation        results unchanged), or ideally the feature selection would be 
 performed in each fold of the CV (which *may* potentially lead to a slightly better external validation, if indeed the hyperparameters led to a poorly  generalizable model).

 2. Clinical features such as age, CA-125, family history etc. could be easily included in the radiomics models and help boost the performance. 

 3. Line 121: The manuscript says that the "primary tumor" was segmented. Was the primary site always the ovaries or pelvis, or was there any            variability that could confound the results?

 4. Lines 251 ff. : It would be useful to use the full names for the radiomic features, including the family, as the used names are somewhat cryptic     (what is F_cm and what is the difference with respect to F_cm_merged, for example?).

 5. Line 227: Does the bootstrap resampling refer to the patients in the test set? Please clarify.

 6. There are some typos and spelling mistakes that could be fixed with a spell-checker.

Author Response

Thank you for the time you dedicated to our paper and for the valuable suggestions you gave us.

You will find our point-to-point responses to your review in the attached file.

Reviewer 2 Report

The authors present ovarian cancer models based on CT images from various oncological centers to predict BRCA mutation and relapse.
- Since the multicentric aspect depicts the novelty or focus of this work, this could be shortly detailed already in the abstract. 
- They conclude that for models to be more accurate the input has to be harmonized. Regarding this seemingly central message of this manuscript, the introduction should also summarize this topic. 
Hereby, I would suggest to move the introduction to Combat harmonization from the methods section up and further elaborate on state of the art approaches.
- Moreover, modeling approaches to radiomics in general and ovarian cancer could be also summarized in the introduction in more detail.
- Models could be further investigated in some more detail by e.g. using an explainability methods.
- In terms of reproducibility, though datasets could be provided anonymized and centralized, still, all underlying code could be provided using Github or other resources.

Author Response

(The authors gave the same response as above.)

Reviewer 3 Report

Dear Authors,

Thank you very much for allowing me to express my opinions related to your work. As a researcher myself, I admire and respect the effort you put into constructing your study and building this manuscript.

Bellow, you can find my comments regarding certain issues. I hope these comments will help you improve both your current and future work.

Abstract

Introduction

  • Lines 77-78: Probably the most important and most evaluated role of radiomics in ovarian pathologies is the differentiation between benign and malignant entities – please address this matter (10.3390/diagnostics11050812, DOI: 10.3390/jpm10030127)
  • Line 82: I will also add that (probably most) of the previous studies included a small number of subjects (and had a pilot nature). This will enforce the strengths of your manuscript.

Materials and Method

  • Choice of imaging method: CT is crucial for OC’s preoperative staging. However, MRI is the most important tool for characterizing these lesions. Moreover, the MRI images are more heterogenous (and contain more “texture information”) compared to the CT images (which have a relatively low tissue contrast), at it was previously-theorized (DOI: 10.1016/j.ejrad.2016.01.013). Therefore, please justify the choice for choosing CT examinations and acknowledge this as a possible limitation of your study.
  • Lines 107-110: You must specify the main acquisition parameters and the machines that acquired your examinations.
  • Line 123: please better define what you refer to as “Gross Tumor 123 Volume”
  • Lines 121-2: The radiomics workflow differs from group 1 [refference 16] compared to groups 2,3,4 in terms of CT machine (presumptively), software and method used for VOI definition, number of texture parameters (group 1 – 1419 parameters, groups 2,3,4 – 217 features), subsequent analysis, etc. I am sure you are aware that TA parameters are very sensitive to the slightest differences in the acquisition and processing protocols, nevertheless to the extraction software (which may use a different method for calculation). You provide brief explanations in paragraphs 157-175 – however, from my POV, it would have made more sense to re-segment the CT tumors from group 1 with the same software you analyzed group 2,3,4 (*RStudio), instead of applying several “statical corrections”. What is your opinion on this? Also, by the matter you proposed, you ignored 1202 features (1419-217) that were extracted from group 1 and not from the other groups.
  • Radiomic and deep learning models – please consider summarising these procedures with a workflow diagram.

Results

  • Table 1 is not written in the same font as the rest of the manuscript’s text. The same issue is with table 2 and Table 3.
  • Figure 2 is very low detailed. Consider splitting this figure into two different figures with higher resolution. The font is not like the rest of the manuscript’s text. Some acronyms within the diagrams are not explained in the figure’s footer.
  • The acronyms regarding features’ names are not explained within the text.
  • You performed ROC analysis but yet no ROC curve was displayed.

 Discussion

  • Lines 304-311: This paragraph repeats the information that was already found in the introduction.
  • Line 320: “CT scan is a more standardized 319 imaging modality, less subject to variability compared to US and MRI.” – please provide an adequate citation for this affirmation.
  • Line 320: “US” and “MRI” acronyms were not explained.
  • Paragraph 318-322: would rather include it as a limitation.
  • Paragraph 323-328: again, this information is more suitable for the introduction. Also, please provide adequate citations.
  • Paragraph between lines 338-374: Comparing your results with other studies needs to be more extensive. Try to find imaging or histopathological explanations for the parameters that had statistically significant results. Also, it is missing is the fluency between paragraphs.
  • Line 392: “..suggesting lack of information in original images.” – please provide more details about how you came to this conclusion.
  • Line 431: why do you consider a limitation the fact that you did not include those characteristics? For a radiology paper, I think you fitted the aim, so you can remove this from the limitations section

Conclusion

  • The conclusion needs to be restrained to one paragraph and a maximum of three short phrases.

Thank you very much for allowing me to express my opinions.

With gratitude,

P.

Author Response

(The authors gave the same response as above.)

Round 2

Reviewer 1 Report

I have no further comments.

Reviewer 3 Report

Thank you for your prompt replies. I believe the new version of the manuscript has an increased scientific value.